# Learning Robust Global Representations
# by Penalizing Local Predictive Power

**Haohan Wang, Songwei Ge, Eric P. Xing, Zachary C. Lipton**
School of Computer Science
Carnegie Mellon University
Pittsburgh, PA 15213
{haohanw,songweig,epxing,zlipton}@cs.cmu.edu

## Abstract

Despite their well-documented predictive power on i.i.d. data, convolutional neural networks have been demonstrated to rely more on high-frequency (textural) patterns that humans deem superficial than on low-frequency patterns that agree better with intuitions about what constitutes category membership. This paper proposes a method for training robust convolutional networks by penalizing the predictive power of the local representations learned by earlier layers. Intuitively, our networks are forced to discard predictive signals such as color and texture that can be gleaned from local receptive fields and to rely instead on the global structure of the image. Across a battery of synthetic and benchmark domain adaptation tasks, our method confers improved generalization. To evaluate cross-domain transfer, we introduce ImageNet-Sketch, a new dataset consisting of sketch-like images and matching the ImageNet classification validation set in categories and scale.

## 1 Introduction

Consider the task of determining whether a photograph depicts a *tortoise* or a *sea turtle*. A human might check to see whether the shell is dome-shaped (indicating tortoise) or flat (indicating turtle). She might also check to see whether the feet are short and bent (indicating tortoise) or fin-like and webbed (indicating turtle). However, the pixels corresponding to the turtle (or tortoise) itself are not alone in offering predictive value. As easily confirmed through a Google Image search, sea turtles tend to be photographed in the sea while tortoises tend to be photographed on land.

Although an image's background may indeed be predictive of the category of the depicted object, it nevertheless seems unsettling that our classifiers should depend so precariously on a signal that is in some sense *irrelevant*. After all, a tortoise appearing in the sea is still a tortoise and a turtle on land is still a turtle. One reason why we might seek to avoid such a reliance on correlated but semantically unrelated artifacts is that they might be liable to change out-of-sample. Even if all cats in a training set appear indoors, we might require a classifier capable of recognizing an *outdoors cat* at test time. Indeed, recent papers have attested to the tendency of neural networks to rely on *surface statistical regularities* rather than learning global concepts (Jo and Bengio, 2017; Geirhos et al., 2019). A number of papers have demonstrated unsettling drops in performance when convolutional neural networks are applied to out-of-domain testing data, even in the absence of adversarial manipulation.

The problem of developing robust classifiers capable of performing well on out-of-domain data is broadly known as Domain Adaptation (DA). While the problem is known to be impossible absent any restrictions on the relationship between training and test distributions (Ben-David et al., 2010b), progress is often possible under reasonable assumptions. Theoretically-principled algorithms have been proposed under a variety of assumptions, including covariate shift (Shimodaira, 2000; Gretton et al., 2009) and label shift (Storkey, 2009; Schölkopf et al., 2012; Zhang et al., 2013; Lipton et al.,

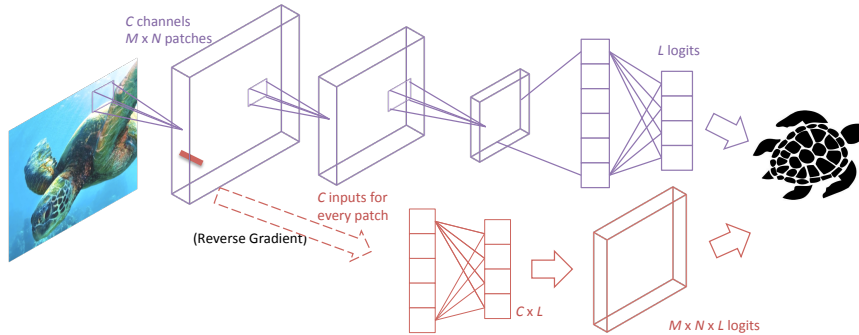

Figure 1: In addition to the primary classifier, our model consists of a number of side classifiers, applied at each $1 \times 1$ location in a designated early layer. The side classifiers result in one prediction per spatial location. The goal of **patch-wise adversarial regularization** is to fool all of them (via reverse gradient) while nevertheless outputting the correct class from the topmost layer.

2018). Despite some known impossibility results for general DA problems (Ben-David et al., 2010b), in practice, humans exhibit remarkable robustness to a wide variety of distribution shifts, exploiting a variety of invariances, and knowledge about what a label actually means.

Our work is motivated by the intuition that for the classes typically of interest in many image classification tasks, the larger-scale structure of the image is what *makes* the class apply and while small local patches might be predictive of the label, Such local features, considered independently, *should not* (vis-a-vis robustness desiderata) comprise the basis for outputting a given classification. Instead, we posit that classifiers that are required to (in some sense) discard this local signal (i.e., patches of an images correlated to the label within a data collection), basing predictions instead on global concepts (i.e., concepts that can only be derived by combining information intelligently across regions), may better mimic the robustness that humans demonstrate in visual recognition.

In this paper, in order to coerce a convolutional neural network to focus on the global concept of an object, we introduce **Patch-wise Adversarial Regularization (PAR)**, a learning scheme that penalizes the predictive power of local representations in earlier layers. The method consists of a patch-wise classifier applied at each spatial location in low-level representation. Via the reverse gradient technique popularized by Ganin et al. (2016), our network is optimized to *fool* the side classifiers and simultaneously optimized to output correct predictions at the final layer. Design choices of PAR include the layer on which the penalty is applied, the regularization strength, and the number of layers in the patch-wise network-in-network classifier.

In extensive experiments across a wide spectrum of synthetic and real data sets, our method outperforms the competing ones, especially when the domain information is not available. We also take measures to evaluate our model's ability to learn concepts at real-world scale despite the small scale of popular domain adaptation benchmarks. Thus we introduce a new benchmark dataset that resembles ImageNet in the choice of categories and size, but consists only of images with the aesthetic of hand-drawn sketches. Performances on this new benchmark also endorse our regularization.

## 2 Related Work

A broad set of papers have addressed various formulations of DA (Bridle and Cox, 1991; Ben-David et al., 2010a) dating in the ML and statistics literature to early works on covariate shift Shimodaira (2000) with antecedents classic econometrics work on sample selection bias (Heckman, 1977; Manski and Lerman, 1977). Several modern works address principled learning techniques under covariate shift (when $p(y|x)$ does not change) (Gretton et al., 2009) and under label shift (when $p(x|y)$ doesn't change) (Storkey, 2009; Zhang et al., 2013; Lipton et al., 2018), and various other assumptions (e.g. bounded divergences between source and target distributions) (Mansour et al., 2009; Hu et al., 2016).

With the recent success of deep learning methods, a number of heuristic domain adaptation methods have been proposed that despite lacking theoretical backing nevertheless confer improvements on a number of benchmarks, even when traditional assumptions break down (e.g., no shared support).

At a high level these methods comprise two subtypes: fine-tuning over target domain (Long et al., 2016; Hoffman et al., 2017; Motiian et al., 2017a; Gebru et al., 2017; Volpi et al., 2018) and coercing domain invariance via adversarial learning (or further extensions) (Ganin et al., 2016; Bousmalis et al., 2017; Tzeng et al., 2017; Xie et al., 2018; Hoffman et al., 2018; Long et al., 2018; Zhao et al., 2018b; Kumar et al., 2018; Li et al., 2018b; Zhao et al., 2018a; Schoenauer-Sebag et al., 2019). While some methods have justified domain-adversarial learning by appealing to theoretical bounds due to Ben-David et al. (2010a), the theory does not in fact guarantee generalization (recently shown by Johansson et al. (2019) and Wu et al. (2019)) and sometimes guarantees failure. For a general primer, we refer to several literature reviews (Weiss et al., 2016; Csurka, 2017; Wang and Deng, 2018).

In contrast to the typical unsupervised DA setup, which requires access to both labeled source data and unlabeled target data, several recent papers propose deep learning methods that confer robustness to a variety of natural-seeming distribution shifts (in practice) without requiring any data (even unlabeled data) from the target distribution. In domain generalization (DG) methods (Muandet et al., 2013) (or sometimes known as "zero shot domain adaptation" (Kumagai and Iwata, 2018; Niu et al., 2015; Erfani et al., 2016; Li et al., 2017c)) one possesses domain identifiers for a number of known in-sample domains, and the goal is to generalize to a new domain. More recent DG approaches incorporate adversarial (or similar) techniques (Ghifary et al., 2015; Wang et al., 2016; Motiian et al., 2017b; Li et al., 2018a; Carlucci et al., 2018), or build ensembles of per-domain models that are then fused representations together (Bousmalis et al., 2016; Ding and Fu, 2018; Mancini et al., 2018). Meta-learning techniques have also been explored (Li et al., 2017b; Balaji et al., 2018).

More recently, Wang et al. (2019) demonstrated promising results on a number of benchmarks without using domain identifiers. Their method achieves addresses distribution shift by incorporating a new component intended to be especially sensitive to domain-specific signals. Our paper extends the setup of (Wang et al., 2019) and empirically studies the problem of developing image classifiers robust to a variety of natural shifts without leveraging any domain information at training or deployment time.

## 3  Method

We use $\langle \mathbf{X}, \mathbf{y} \rangle$ to denote the samples and $f(g(\cdot; \delta); \theta)$ to denote a convolutional neural network, where $g(\cdot; \delta)$ denotes the output of the bottom convolutional layers (e.g., the first layer), and $\delta$ and $\theta$ are parameters to be learned. The traditional training process addresses the optimization problem

$$\min_{\delta, \theta} \mathbb{E}_{(\mathbf{X}, \mathbf{y})}[l(f(g(\mathbf{X}; \delta); \theta), \mathbf{y})], \tag{1}$$

where $l(\cdot, \cdot)$ denotes the loss function, commonly cross-entropy loss in classification problems.

Following the standard set-up of a convolutional layer, $\delta$ is a tensor of $c \times m \times n$ parameters, where $c$ denotes the number of convolutional channels, and $m \times n$ is the size of the convolutional kernel. Therefore, for the $i^{\text{th}}$ sample, $g(\mathbf{X}_i; \delta)$ is a representation of $\mathbf{X}_i$ of the dimension $c \times m' \times n'$, where $m'$ (or $n'$) is a function of the image dimension and $m$ (or $n$). [1]

### 3.1  Patch-wise Adversarial Regularization

We first introduce a new classifier, $h(\cdot; \phi)$ that takes the input of a $c$-length vector and predicts the label. Thus, $h(\cdot; \phi)$ can be applied onto the representation $g(\mathbf{X}_i; \delta)$ and yield $m' \times n'$ predictions. Therefore, each of the $m' \times n'$ predictions can be seen as a prediction made only by considering a small image patch corresponding to each of the receptive fields in $g(\mathbf{X}_i; \delta)$. If any of the image patches are predictive and $g(\cdot; \delta)$ summarizes the predictive representation well, $h(\cdot; \phi)$ can be trained to achieve a high prediction accuracy.

On the other hand, if $g(\cdot; \delta)$ summarizes the patch-wise predictive representation well, higher layers ($f(\cdot; \theta)$) can directly utilize these representation for prediction and thus may not be required to learn a global concept. Our intuition is that by regularizing $g(\cdot; \delta)$ such that each fiber (i.e., representation at the same location from every channel) in the activation tensor should not be individually predictive of the label, we can prevent our model from relying on local patterns and instead force it to learn a pattern that can only be revealed by aggregating information across multiple receptive fields.

As a result, in addition to the standard optimization problem (Eq. 1), we also optimize the following term:

$$\min_{\phi} \max_{\delta} \mathbb{E}_{(\mathbf{X},\mathbf{y})} \big[ \sum_{i,j}^{m',n'} l(h(g(\mathbf{X};\delta)_{i,j};\phi),\mathbf{y}) \big] \qquad (2)$$

where the minimization consists of training $h(\cdot;\phi)$ to predict the label based on the local features (at each spatial location) while the maximization consists of training $g(\cdot;\delta)$ to shift focus away from local predictive representations.

We hypothesize that by jointly solving these two optimization problems (Eq. 1 and Eq. 2), we can train a model that can predict the label well without relying too strongly on local patterns. The optimization can be reformulated into the following two problems:

$$\min_{\delta,\theta} \mathbb{E}_{(\mathbf{X},\mathbf{y})}[l(f(g(\mathbf{X};\delta);\theta),\mathbf{y}) - \frac{\lambda}{m'n'} \sum_{i,j}^{m',n'} l(h(g(\mathbf{X};\delta)_{i,j};\phi),\mathbf{y})]$$

$$\min_{\phi} \mathbb{E}_{(\mathbf{X},\mathbf{y})}[\frac{\lambda}{m'n'} \sum_{i,j}^{m',n'} l(h(g(\mathbf{X};\delta)_{i,j};\phi),\mathbf{y})]$$

where $\lambda$ is a tuning hyperparameter. We divide the loss by $m'n'$ to keep the two terms at a same scale.

Our method can be implemented efficiently as follows: In practice, we consider $h(\cdot;\phi)$ as a fully-connected layer. $\phi$ consists of a $c \times k$ weight matrix and a $k$-length bias vector, where $k$ is the number of classes. The $m' \times n'$ forward operations as fully-connected networks can be efficiently implemented as a $1 \times 1$ convolutional operation with $c$ input channels and $k$ output channels operating on the $m' \times n'$ representation.

Note that although the input has $m' \times n'$ vectors, $h(\cdot;\phi)$ only has one set of parameters that is used for all these vectors, in contrast to building a set of parameter for every receptive field of the $m' \times n'$ dimension. Using only one set of parameters can not only help to reduce the computational load and parameter space, but also help to identify the predictive local patterns well because the predictive local pattern does not necessarily appear at the same position across the images. Our idea of our method is illustrated in Figure 1.

## 3.2 Other Extensions and Training Heuristics

There can be many simple extensions to the basic PAR setting we discussed above. Here we introduce three extensions that we will experiment with later in the experiment section.

**More Powerful Pattern Classifier:** We explore the space of discriminator architectures, replacing the single-layer network $h(\cdot;\phi)$ with a more powerful network architecture, e.g. a multilayer perceptron (MLP). In this paper, we consider three-layer MLPs with ReLU activation functions. We name this variant as PAR$_\text{M}$.

**Broader Local Pattern:** We can also extend the $1 \times 1$ convolution operation to enlarge the concept of "local". In this paper, we experiment with a $3 \times 3$ convolution operation, thus the number of parameters in $\phi$ is increased. We refer to this variant as PAR$_\text{B}$.

**Higher Level of Local Concept:** Further, we can also build the regularization upon higher convolutional layers. Building the regularization on higher layers is related to enlarging the patch of image, but also considering higher level of abstractions. In this paper, we experiment the regularization on the second layer. We refer this method as PAR$_\text{H}$.

**Training Heuristics:** Finally, we introduce the training heuristic that plays an important role in our regularization technique, especially in modern architectures such as AlexNet or ResNet. The training heuristic is simple: we first train the model conventionally until convergence (or after a certain number of epochs), then train the model with our regularization. In other words, we can also directly work on pretrained models and continue to fine-tune the parameters with our regularization.

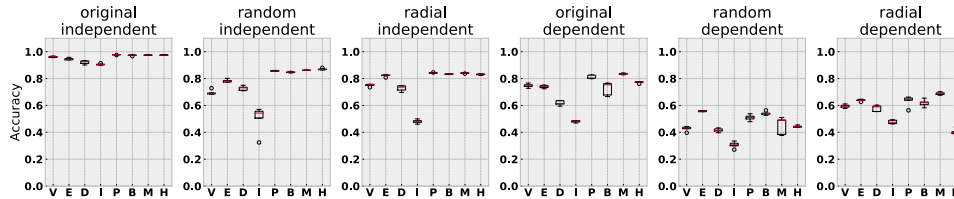

Figure 2: Prediction accuracy with standard deviation for MNIST with patterns. Notations: V: vanilla baseline, E: HEX, D: DANN, I: InfoDrop, P: PAR, B: PAR$_B$, M: PAR$_M$, H: PAR$_H$

# 4 Experiments

In this section, we test PAR over a variety of settings, we first test with perturbed MNIST under the domain generalization setting, and then test with perturbed CIFAR10 under domain adaptation setting. Further, we test on more challenging data sets, with PACS data under domain generalization setting and our newly proposed ImageNet-Sketch data set. We compare with previous state-of-the-art when available, or with the most popular benchmarks such as DANN (Ganin et al., 2016), InfoDrop (Achille and Soatto, 2018), and HEX (Wang et al., 2019) on synthetic experiments.[2][3]

## 4.1 MNIST with Perturbation

We follow the set-up of Wang et al. (2019) in experimenting with MNIST data set with different superficial patterns. There are three different superficial patterns (radial kernel, random kernel, and original image). The training/validation samples are attached with two of these patterns, while the testing samples are attached with the remaining one. As in Wang et al. (2019), training/validation samples are attached with patterns following two strategies: 1) *independently*: the pattern is independent of the digit, and 2) *dependently*: images of digit 0-4 have one pattern while images of digit 5-9 have the other pattern.

We use the same model architecture and learning rate as in Wang et al. (2019). The extra hyperparameter $\lambda$ is set as 1 as the most straightforward choice. Methods in Wang et al. (2019) are trained for 100 epochs, so we train the model for 50 epochs as pretraining and 50 epochs with our regularization. The results are shown in Figure 2. In addition to the direct message that our proposed method outperforms competing ones in most cases, it is worth mentioning that the proposed methods behave differently in the "dependent" settings. For example, PAR$_M$ performs the best in the "original" and "radial" settings, but almost the worst among proposed methods in the "random" setting, which may indicate that the pattern attached by "random" kernel can be more easily detected and removed by PAR$_M$ during training (Notice that the name of the setting ("original", "radial" or "random") indicates the pattern attached to testing images, and the training samples are attached with the other two patterns). More information about hyperparameter choice is in Appendix A.

## 4.2 CIFAR with Perturbation

We continue to experiment on CIFAR10 data set by modifying the color and texture of test dataset with four different schemas: 1) greyscale; 2) negative color; 3) random kernel; 4) radial kernel. Some examples of the perturbed data are shown in Appendix B. In this experiment, we use ResNet-32 as our base classifier, which has a rough 92% prediction accuracy on original CIFAR10 test data set.

As for PAR, we first train the base classifier for 250 epochs and then train with the adversarial loss for another 150 epochs. As for the competing models, we also train for 400 epochs with carefully selected hyperparameters. The overall performances are shown in Table 1. In general, PAR and its variants achieve the best performances on all four test data sets, even when DANN has an unfair advantage over others by seeing unlabelled testing data during training. To be specific, PAR achieves the best performances on the greyscale and radial kernel settings; PAR$_M$ is the best on the negative color and random kernel settings. One may argue that the numeric improvements are not significant

Table 1: Test accuracy of PAR and variants on Cifar10 datasets with perturbed color and texture.

|  | ResNet | DANN | InfoDrop | HEX | PAR | $PAR_B$ | $PAR_M$ | $PAR_H$ |
|---|---|---|---|---|---|---|---|---|
| Greyscale | 87.7 | 87.3 | 86.4 | 87.6 | **88.1** | 87.9 | 87.8 | 86.9 |
| NegColor | 62.8 | 64.3 | 57.6 | 62.4 | 66.2 | 65.3 | **67.6** | 62.7 |
| RandKernel | 43.0 | 33.4 | 41.3 | 42.5 | 47.0 | 40.5 | **47.5** | 40.8 |
| RadialKernel | 62.4 | 63.3 | 60.3 | 61.9 | **63.8** | 63.2 | 63.2 | 61.4 |
| Average | 63.9 | 62.0 | 61.4 | 63.6 | 66.3 | 64.2 | **66.5** | 62.9 |

and PAR may only affect the model marginally, but a closer look at the training process of the methods indicates that our regularization of local patterns benefits the robustness significantly while minimally impacting the original performance. More detailed discussions are in Appendix B.

## 4.3 PACS

We test on the PACS data set (Li et al., 2017a), which consists of collections of images over four domains, including photo, art painting, cartoon, and sketch. Many recent methods have been tested on this data set, which offers a convenient way for PAR to be compared with the previous state-of-the-art. Following Li et al. (2017a), we use AlexNet as baseline and build PAR upon it. We compare with recently reported state-of-the-art on this data set, including DSN (Bousmalis et al., 2016), LCNN (Li et al., 2017a), MLDG (Li et al., 2017b), Fusion (Mancini et al., 2018), MetaReg (Balaji et al., 2018), Jigen (Carlucci et al., 2019), and HEX (Wang et al., 2019), in addition to the baseline reported in (Li et al., 2017a). We are also aware that methods that explicitly use domain knowledge (*e.g.*, Lee et al., 2018) may be helpful, but we do not directly compare with them numerically, as the methods deviate from the central theme of this paper.

Table 2: Prediction accuracy of PAR and variants on PACS data set in comparison with the previously reported state-of-the-art results. Bold numbers indicate the best performance (three sets, one for each scenario). We use $\star$ to denote the methods that use the training setting in (Carlucci et al., 2019) (*e.g.*, extra data augmentation, different train-test split, and different learning rate scheduling). Notably, $PAR_H$ achieves the best performance in sketch testing case even in comparison to all other methods without data augmentation.

|  | Art | Cartoon | Photo | Sketch | Average | Forgoing Domaim ID | Data Aug. |
|---|---|---|---|---|---|---|---|
| AlexNet | 63.3 | 63.1 | 87.7 | 54 | 67.03 | ✓ |  |
| DSN | 61.1 | 66.5 | 83.2 | 58.5 | 67.33 |  |  |
| L-CNN | 62.8 | 66.9 | 89.5 | 57.5 | 69.18 |  |  |
| MLDG | 63.6 | 63.4 | 87.8 | 54.9 | 67.43 |  |  |
| Fusion | 64.1 | 66.8 | 90.2 | **60.1** | 70.30 |  |  |
| MetaReg | **69.8** | **70.4** | **91.1** | 59.2 | **72.63** |  |  |
| HEX | 66.8 | 69.7 | 87.9 | 56.3 | 70.18 | ✓ |  |
| PAR | **66.9** | 67.1 | 88.6 | 62.6 | 71.30 | ✓ |  |
| $PAR_B$ | 66.3 | 67.8 | 87.2 | 61.8 | 70.78 | ✓ |  |
| $PAR_M$ | 65.7 | 68.1 | 88.9 | 61.7 | 71.10 | ✓ |  |
| $PAR_H$ | 66.3 | **68.3** | **89.6** | **64.1** | **72.08** | ✓ |  |
| Jigen$\star$ | 67.6 | **71.7** | 89.0 | **65.1** | 73.38 | ✓ | ✓ |
| PAR$\star$ | 68.0 | 71.6 | **90.8** | 61.8 | 73.05 | ✓ | ✓ |
| $PAR_B\star$ | 67.6 | 70.7 | 90.1 | 62.0 | 72.59 | ✓ | ✓ |
| $PAR_M\star$ | **68.7** | 71.5 | 90.5 | 62.6 | 73.33 | ✓ | ✓ |
| $PAR_H\star$ | **68.7** | 70.5 | 90.4 | 64.6 | *73.54* | ✓ | ✓ |

Following the training heuristics we introduced, we continue with trained AlexNet weights[4] and fine-tune on training domain data of PACS for 100 epochs. We notice that once our regularization is plugged in, we can outperform the baseline AlexNet with a 2% improvement. The results are

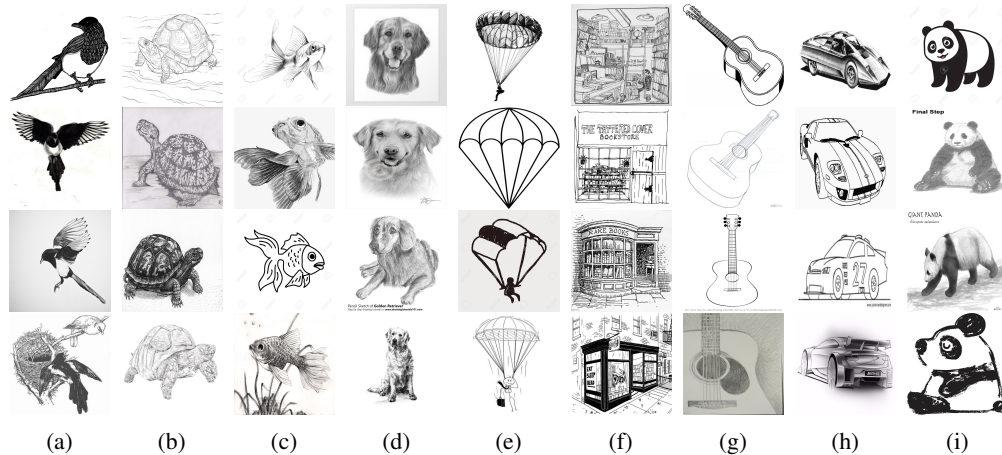

(a)  (b)  (c)  (d)  (e)  (f)  (g)  (h)  (i)

Figure 3: Sample Images from ImageNet-Sketch. Corresponding classes: (a) magpie (b) box turtle (c) goldfish (d) golden retriever (e) parachute (f) bookshop (g) acoustic guitar (h) racer (i) giant panda

reported in Table 2, where we separate the results of techniques relying on domain identifications and techniques free of domain identifications.

We also report the results based on the training schedule used by (Carlucci et al., 2019) as shown in the bottom part of Table 2. Note that (Carlucci et al., 2019) used the random training-test split that are different from the official split used by the other baselines. In addition, they used another data augmentation technique to convert image patch to grayscale which could benefit the adaptation to Sketch domain.

While our methods are in general competitive, it is worth mentioning that our methods improve upon previous methods with a relatively large margin when Sketch is the testing domain. The improvement on Sketch is notable because Sketch is the only colorless domain out of the four domains in PACS. Therefore, when tested with the other three domains, a model may learn to exploit the color information, which is usually local, to predict, but when tested with Sketch domain, the model has to learn colorless concepts to make good predictions.

## 4.4 ImageNet-Sketch

### 4.4.1 The ImageNet-Sketch Data

Inspired by the Sketch data of (Li et al., 2017a) with seven classes, and several other Sketch datasets, such as the *Sketchy* dataset (Sangkloy et al., 2016) with 125 classes and the *Quick Draw!* dataset (QuickDraw, 2018) with 345 classes, and motivated by absence of a large-scale sketch dataset fitting the shape and size of popular image classification benchmarks, we construct the ImageNet-Sketch data set for evaluating the out-of-domain classification performance of vision models trained on ImageNet.

Compatible with standard ImageNet validation data set for the classification task (Deng et al., 2009), our ImageNet-Sketch data set consists of 50000 images, 50 images for each of the 1000 ImageNet classes. We construct the data set with Google Image queries "sketch of _____", where _____ is the standard class name. We only search within the "black and white" color scheme. We initially query 100 images for every class, and then manually clean the pulled images by deleting the irrelevant images and images that are for similar but different classes. For some classes, there are less than 50 images after manually cleaning, and then we augment the data set by flipping and rotating the images.

We expect ImageNet-Sketch to serve as a unique ImageNet-scale out-of-domain evaluation dataset for image classification. Also, notably, different from perturbed ImageNet validation sets (Geirhos et al., 2019; Hendrycks and Dietterich, 2019), the images of ImageNet-Sketch are collected independently from the original validation images. The independent collection procedure is more similar to (Recht et al., 2019), who collected a new set of standard colorful ImageNet validation images. However, while their goal was to assess overfitting to the benchmark validation sets, and thus they tried replicate

Table 3: Testing accuracy of competing methods on the ImageNet-Sketch data. The bottom half denotes the method that has extra advantages: † denotes the method that has access to unlabelled target domain data, and ⋆ denotes the method that use extra data augmentation.

|  | AlexNet | InfoDrop | HEX | PAR | PAR$_B$ | PAR$_M$ | PAR$_H$ |
|---|---|---|---|---|---|---|---|
| Top 1 | 0.1204 | 0.1224 | 0.1292 | **0.1306** | 0.1273 | 0.1287 | 0.1266 |
| Top 5 | 0.2480 | 0.2560 | **0.2654** | 0.2627 | 0.2575 | 0.2603 | 0.2544 |
|  | DANN† | JigGen⋆ | PAR⋆ | PAR$_B$⋆ | PAR$_M$⋆ | PAR$_H$⋆ |  |
| Top 1 |  | 0.1360 | 0.1469 | 0.1494 | 0.1494 | **0.1501** | 0.1499 |
| Top 5 |  | 0.2712 | 0.2898 | 0.2949 | 0.2945 | **0.2957** | 0.2954 |

Table 4: Some examples that are predicted correctly with our method but wrongly with the original AlexNet because the original model seems to focus on the local patterns.

|  | AlexNet-PAR | | AlexNet | |
|---|---|---|---|---|
|  | prediction | confidence | prediction | confidence |
| 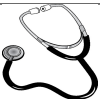 | stethoscope | 0.6608 | hook | 0.3903 |
|  | tricycle | 0.9260 | safety pin | 0.5143 |
|  | Afghan hound | 0.8945 | swab (mop) | 0.7379 |
|  | red wine | 0.5999 | goblet | 0.7427 |

the ImageNet collection procedure exactly, our goald is to collect out-of-domain black-and-white sketch images with the goal of testing a model's ability to extrapolate out of domain.[5] Sample images are shown in Figure 3.

### 4.4.2 Experiment Results

We use AlexNet as the baseline and test whether our method can help improve the out-of-domain prediction. We start with ImageNet pretrained AlexNet and continue to use PAR to tune AlexNet for another five epochs on the original ImageNet training dataset. The results are reported in Table 3.

We are particularly interested in how PAR improves upon AlexNet, so we further investigate the top-1 prediction results. Although the numeric results in Table 3 seem to show that PAR only improves the upon AlexNet by predicting a few more examples correctly, we notice that these models share 5025 correct predictions, while AlexNet predicts another 1098 images correctly and PAR predicts a different set of 1617 images correctly.

We first investigate the examples that are correctly predicted by the original AlexNet, but wrongly predicted by PAR. We notice some examples that help verify the performance of PAR. For examples, PAR incorrectly predicts three instances of "keyboard" as "crossword puzzle," while AlexNet predicts these samples correctly. It is notable that two of these samples are "keyboards with missing keys" and hence look similar to a "crossword puzzle."

We also investigate the examples that are correctly predicted by PAR, but wrongly predicted by the original AlexNet. Interestingly, we notice several samples that are wrongly predicted by AlexNet because the model may only focus on the local patterns. Some of the most interesting examples are reported in Table 4: The first example is a stethoscope, PAR predicts it correctly with 0.66 confidence, while AlexNet predicts it to be a hook. We conjecture the reason to be that AlexNet tends to only focus on the curvature which resembles a hook. The second example tells a similar story, PAR

predicts tricycle correctly with 0.92 confidence, but AlexNet predicts it as a safety pin with 0.51 confidence. We believe this is because part of the image (likely the seat-supporting frame) resembles the structure of a safety pin. For the third example, PAR correctly predicts it to be an Afghan hound with 0.89 confidence, but AlexNet predicts it as a mop with 0.73 confidence. This is likely because the fur of the hound is similar to the head of a mop. For the last example, PAR correctly predicts the object to be red wine with 0.59 confidence, but AlexNet predicts it to be a goblet with 0.74 confidence. This is likely because part of the image is indeed part of a goblet, but PAR may learn to make predictions based on the global concept considering the bottle, the liquid, and part of the goblet together. Table 4 only highlights a few examples, and more examples are shown in Appendix C.

## 5   Conclusion

In this paper, we introduced *patch-wise adversarial regularization*, a technique that regularizes models, encouraging them to learn *global concepts* for classifying objects by penalizing the model's ability to make predictions based on representations of *local* patches. We extended our basic set-up with several different variants and conducted extensive experiments, evaluating these methods with several datasets for domain adaptation and domain generalization tasks. The experimental results favored our methods, especially when domain information is unknown to the methods. In addition to the superior performances we achieved through these experiments, we expected to further challenge our method at real-world scale. Therefore, we also constructed a dataset that matches the ImageNet classification validation set in classes and scales but contains only sketch-alike images. Our new ImageNet-Sketch data set can serve as new territory for evaluating models' ability to generalize to out-of-domain images at an unprecedented scale.

While our method often confers benefits on out-of-domain data, we note that it may not help (or can even hurt) in-domain accuracy when local patterns are truly predictive of the labels. However, we argue that the local patterns, while predictive in-sample, may be less reliable out-of-domain as compared to larger-scale patterns, which motivates this paper. For the three variations we introduced, our experiments indicate that different variants are applicable to different scenarios. We recommend that users decide which variant to use given their understanding of the problem and hope in future work, to develop clear principles for guiding these choices. While we did not give a clear choice of which PAR to use, we note that none of the variants of PAR outperform the vanilla PAR consistently. However, the vanilla PAR outperforms most comparable baselines in the vast majority of our experiments.

### Acknowledgments

Haohan Wang is supported by NIH R01GM114311, NIH P30DA035778, and NSF IIS1617583. Any opinions, findings and conclusions or recommendations expressed in this material are those of the author(s) and do not necessarily reflect the views of the National Institutes of Health or the National Science Foundation. Zachary Lipton thanks the Center for Machine Learning and Health, a joint venture of Carnegie Mellon University, UPMC, and the University of Pittsburgh for supporting our collaboration with Abridge AI to develop robust models for machine learning in healthcare. He is also grateful to Salesforce Research, Facebook Research, and Amazon AI for faculty awards supporting his lab's research on robust deep learning under distribution shift.

## Footnotes

[1]The exact function depends on padding size and stride size, and is irrelevant to the discussion of this paper.

[2]Clean demonstration of the implementation can be found at: https://github.com/HaohanWang/PAR

[3]Source code for replication can be found at : https://github.com/HaohanWang/PAR_experiments

[4]https://www.cs.toronto.edu/~guerzhoy/tf_alexnet/

[5]The ImageNet-Sketch data can be found at: https://github.com/HaohanWang/ImageNet-Sketch

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
