[Supplementary Material]



Figure 4: Prediction accuracy with standard deviation for MNIST with superficial statistics perturbation data set. Notations: V: vanilla baseline, A: PAR with $\lambda = 0.01$, B: PAR with $\lambda = 0.1$, C: PAR with $\lambda = 1$, D: PAR with $\lambda = 10$, E: PAR with $\lambda = 100$

## A  Other Hyperparameter Choices for MNIST experiment

We also experimented with the parameter choices of the method in the MNIST experiment. We varied the $\lambda$ in $\{0.01, 0.1, 1, 10, 100\}$ in PAR and reported the performance to guide further usage of the method.

As we can see from Figure 4, PAR seems to prefer the cases when $\lambda$ is relatively smaller, although what we reported in the main manuscript for the MNIST experiment is $\lambda = 1$ as the most straightforward choice, to demonstrate the method's strength.

Later in other experiments, especially the ImageNet-Sketch experiment, we notice that $\lambda = 1$ is too strong (unless the learning rate is set to be much smaller) for the method to work. We observe that a too-strong $\lambda$ usually immediate deteriorates the performance during first epoches of training. Therefore, in practice, we recommend the users to set the $\lambda$ (or learning rate) to be smaller if the users observe that our method deteriorates the training performance.

# B Cifar10 discussion

(a) Radial mask      (b) Random mask

Figure 5: Fourier filtering kernel

(a) Original      (b) Negative      (c) Greyscale      (d) RandomKernel      (e) RadialKernel

Figure 6: Examples of Cifar10 images with perturbed color and texture.

(a) Layer 1

(b) Layer 2

(c) Layer 3

(d) Layer 4

Figure 7: Prediction accuracy of patch-wise classifier. The regularization is introduced at Epoch 250. we ran experiments to validate the patch-wise classifier. Without (PAR) regularization, the patch-wise classifier can achieve roughly 20% accuracy on in-domain test data ((a), orange, before epoch 250). It achieves 12% accuracy on texture-altered out-of-domain data ((a), magenta and green, before epoch 250) and 5% accuracy color-altered out-of-domain data ((a), maroon, before epoch 250). With PAR, the patch-wise classifier achieves 15% in-domain prediction accuracy (5% drop) ((a), orange, after epoch 250), and 10% on texture-altered out-of-domain data ((a), magenta and green, after epoch 250) and 8% on color-altered out-of-domain data ((a), maroon, after epoch 250).

(a) level 1                  (b) level 2

(c) level 3                  (d) level 4

Figure 8: The solid lines represent the test accuracy of PAR with different levels of local patterns during the training process. For a better comparison, we use the dashed lines to represent the test accuracy at 250 epoch when the adversarial training is firstly added. The default PAR is shown in (a), we can see a small jitter after 250 epoch when the model is coerced to forget the information of local patterns. Then the performances on the perturbed dataset start to increase while the performance on the original dataset is not greatly impacted. In addition, when higher level of local patterns are used, little improvement can be observed, except for using level 2 on the negative color.

Figure 9: Evaluation with different sizes of convolutional filters. Note that all the local pattern classifiers contain one layer and 10 channels but different filter sizes. In general, the performances with different filter sizes on the test datasets are very similar except for $RandomKernel$.

(a) $\lambda = 0.1$

(b) $\lambda = 0.2$

(c) $\lambda = 0.5$

(d) $\lambda = 1.0$

Figure 10: Evaluation on adversarial training with multiple levels and the different decays. Note that all the layers are used for extracting local concepts with a decay, i.e. adding weights $1$, $\lambda$, $\lambda^2$ and $\lambda^3$ to the adversarial losses of four layers. Smaller decay (larger $\lambda$) leads to unstable performances.

Table 5: More prediction comparisions between AlexNet-PAR and AlexNet

| | AlexNet-PAR | | AlexNet | | | AlexNet-PAR | | AlexNet | |
|---|---|---|---|---|---|---|---|---|---|
| Image | Prediction | Conf. | Prediction | Conf. | Image | Prediction | Conf. | Prediction | Conf. |
| | Afghan hound | 0.89 | swab (mop) | 0.74 | | sunglass | 0.42 | strainer | 0.27 |
| | Afghan hound | 0.92 | swab (mop) | 0.82 | | sunglass | 0.31 | strainer | 0.19 |
| | Afghan hound | 0.80 | swab (mop) | 0.20 | | sunglass | 0.38 | strainer | 0.32 |
| | bull mastiff | 0.42 | shower cap | 0.23 | | totem pole | 0.30 | envelope | 0.39 |
| | bull mastiff | 0.33 | shower cap | 0.37 | | totem pole | 0.43 | envelope | 0.27 |
| | bull mastiff | 0.57 | shower cap | 0.77 | | totem pole | 0.50 | envelope | 0.40 |
| | ashcan | 0.17 | safety pin | 0.41 | | totem pole | 0.45 | envelope | 0.39 |
| | ashcan | 0.38 | safety pin | 0.26 | | tricycle | 0.17 | safety pin | 0.42 |
| | ashcan | 0.16 | safety pin | 0.53 | | tricycle | 0.66 | safety pin | 0.49 |
| | car mirror | 0.42 | buckle | 0.89 | | tricycle | 0.07 | safety pin | 0.12 |
| | car mirror | 0.57 | buckle | 0.43 | | tricycle | 0.93 | safety pin | 0.51 |
| | car mirror | 0.88 | buckle | 0.63 | | whiskey jug | 0.20 | perfume | 0.14 |
| | stethoscope | 0.46 | hook | 0.37 | | whiskey jug | 0.65 | perfume | 0.37 |
| | stethoscope | 0.58 | hook | 0.55 | | whiskey jug | 0.49 | perfume | 0.51 |
| | stethoscope | 0.77 | hook | 0.58 | | head cabbage | 0.44 | shower cap | 0.31 |
| | stethoscope | 0.75 | hook | 0.42 | | head cabbage | 0.78 | shower cap | 0.79 |
| | stethoscope | 0.46 | hook | 0.59 | | head cabbage | 0.34 | shower cap | 0.39 |
| | stethoscope | 0.66 | hook | 0.39 | | red wine | 0.32 | goblet | 0.84 |
| | stethoscope | 0.56 | hook | 0.46 | | red wine | 0.60 | goblet | 0.74 |
| | stethoscope | 0.56 | hook | 0.49 | | red wine | 0.82 | goblet | 0.67 |

## C   More results of ImageNet-Sketch

We conducted a detailed analysis of ImageNet-Sketch results with the following rules:

- The samples are correctly predicted by one model, but wrongly predicted by the other.
- When a model makes wrong predictions, the samples in the class tend to be predicted into a same another class. Therefore, we can exclude some random prediction errors.
- For class A and B, if one model tends to predict samples in class A into class B, and the other model has the reverse tendency, we investigate neither of these classes, because the different prediction results may only be due to the similarity of these two classes.

Table 5 shows more samples that are correctly predicted by PAR but wrongly predicted by the original AlexNet, because (as we conjecture) the original AlexNet focuses on local patterns.