[Reviews · NeurIPS 2019]

Reviewer 1



The main argument, that predictions should not be too focused on local patches was a little counter-intuitive. There are many counter examples for the points they made, e.g. a local patch around a person's face may be enough to classify the image as a person. This was a little distracting while reading their paper, but the examples presented were satisfactory to make their points, and the extensive results help alleviate most of my concerns about this. Otherwise, I was happy with the paper, felt it provided an interesting approach for domain adaptation.

Reviewer 2



This paper proposes a new method for learning a robust network by penalizing the local predictive power and forcing it to focus on global features. To penalize the local predictive power, as a regularization, a new network is build upon the features of early layers and adversarial training is adopted to make these early layer features not able to correctly predict. However, there are still some concerns about the method and the experiment. 1. The compared baseline method for domain generalization setting is not state-of-the-art. Some recent methods such as JiGen [1], Feature-Critic [2] and Epi-FCR [3] should be compared. 2. As we can see from the experimental results, the performance of the proposed method and its variants perform differently on different datasets. For MNIST, PAR_M seems to be the best, for CIFAR10, PAR and PAR_M perform better than PAR_B and PAR_H, but for PACS, PAR_H performs best. So how should we select which one to use in realistic applications? 3. In page4, it is described that the method is tested on CIFAR10 under domain adaptation setting, but in section 4.2 it seems for domain generalization. Besides, it is not clear which Resnet is used for the experiments on Cifar10 dataset. 4. The claim that “the proposed network is forced to discard predictive signals such as color and texture” is not well validated. What is the prediction accuracy of the patch-wise classifier? [1] Carlucci F M, D'Innocente A, Bucci S, et al. Domain Generalization by Solving Jigsaw Puzzles[J]. CVPR, 2019. [2] Li Y, Yang Y, Zhou W, et al. Feature-Critic Networks for Heterogeneous Domain Generalization[J]. arXiv preprint arXiv:1901.11448, 2019. [3] Li D, Zhang J, Yang Y, et al. Episodic Training for Domain Generalization[J]. arXiv preprint arXiv:1902.00113, 2019. --------------------- After reading the rebuttal, most of the my concerns are resolved except for missing the recent state-of-the-art baselines. I am leaning towards accepting the paper.

Reviewer 3



we posit that classifiers that are required to (in some sense) discard this local signal (i.e., patches of an images correlated to the label within a data collection), basing predictions instead on global concepts (i.e., concepts that can only be derived by combining information intelligently across regions), may better mimic the robustness that humans demonstrate in visual recognition. I believe the work, together with the work in Wang et al. (2019) in reference, provide a very insight in domain adaptation. The writing is clear, the extensive experimental results make it a solid work. The new dataset provided in the manuscript further benefit the cross-dataset recognition via UDA in computer vision area.

[Author Response · NeurIPS 2019]

**General Reply:**

First, we would like to thank all of the reviewers for their detailed and thoughtful reviews. We are glad to see that reviewers generally appreciated the strengths of our paper, especially the significance of the methodological contribution (R1, R3) including its potential impact in production (R1), the clarity of our writing (R2, R3), and the introduction of the ImageNet-Sketch dataset. Additionally, we thank the reviewers for constructive suggestions, in particular requests for additional analysis to support PAR's effectiveness and modifications to the exposition. Below, we address each reviewer's comments in turn.

**Reply to Reviewer 1**:

*"the main argument was a little counter-intuitive"* We agree that your intuition is right: local patches are often predictive (in-sample), and a large body of evidence suggests that ConvNets naturally exploit this property. Our main argument does not refute this. Instead, we argue that these patches, while predictive in-sample, may be less reliable out-of-domain as compared to larger-scale patterns. We will improve the writing to make this point clearer.

*"how well this approach would work on models focusing on loss functions across multiple layers?"* This is a great question. While we didn't address object detection and indeed, adapting our approach to the SSD architecture might require modifications (say, an additional convolutional layer prepended before the standard SSD layers). However, we see no obstacles to integrating our approach with other object detection architectures, such as Faster R-CNN and YOLO. We are grateful for the suggestions and will explore these directions in future work.

*"comparison to the related work that used domain knowledge"* Thanks for this suggestion. We are working to implement this and other requested baselines and will add the results to the camera-ready version if accepted.

**Reply to Reviewer 2**:

*"missing recent baselines"* Thanks for pointing our these references to recent domain generalization papers. We will add these comparisons to the camera-ready version if accepted.

*"how should we select which PAR to use?"* This is a great question. None of the variants of PAR outperform the vanilla PAR consistently. **However, the vanilla PAR outperforms nearly all other baselines in the vast majority of our experiments.** Our draft includes results for these variants for the sake of thoroughness.

*"clarifications about Section 4.2"* In some experiments, some of our baselines access domain labels (and thus treat the problem as domain generalizaiton). However, our model is blind to domain labels. Moreover, unlike unsupervised domain adaptation approaches, our model does not incorporate the unlabeled target data into training. **Surprisingly, our methods often perform better despite the unfair comparison.** Our experiments employ ResNet-50. We will revised the draft to make these facts clearer and release our code publicly for full transparency.

*"validation of the patch-wise classifier"* Per your suggestion, we ran experiments to validate the patch-wise classifier. Without (PAR) regularization, the patch-wise classifier can achieve roughly 20% accuracy on in-domain test data (Figure 1(a), orange, before epoch 250). It achieves 12% accuracy on texture-altered out-of-domain data (Figure 1(a), magenta and green, before epoch 250) and 5% accuracy color-altered out-of-domain data (Figure 1(a), maroon, before epoch 250). With PAR, the patch-wise classifier achieves 15% in-domain prediction accuracy (5% drop) (Figure 1(a), orange, after epoch 250), and 10% on texture-altered out-of-domain data (Figure 1(a), magenta and green, after epoch 250) and 8% on color-altered out-of-domain data (Figure 1(a), maroon, after epoch 250).

(a) Layer 1      (b) Layer 2      (c) Layer 3      (d) Layer 4

Figure 1: Prediction accuracy of patch-wise classifier. The regularization is introduced at Epoch 250.

**Reply to Reviewer 3**:

Thanks for recognizing the fundamental nature and significance of our contribution.

*"more related theoretical analysis if possible"* We share your enthusiasm for these foundational theoretical results (Ben-David 2010), which have influenced theoretical inquiry into domain adaptation, However, we note that papers claiming theoretical support for deep domain adaptation techniques have misinterpreted the theory. Two recent papers [1,2] independently identified these flaws and construct simple counter-examples (e.g., when label distributions shift) where these techniques are guaranteed to fail (if the optimization succeeds). We agree that theoretical support would be a great asset, noting only that comparable methods, lack such theoretical support.

1. *Johansson et al. Support and Invertibility in Domain-Invariant Representations (AISTATS 2019)*

2. *Wu et al. Domain Adaptation with Asymmetrically-Relaxed Distribution Alignment (ICML 2019)*

[Meta-Review · NeurIPS 2019]

The reviewers appreciated the feedback from the authors and has reached a consensus that the paper should be accepted. Please incorporate the feedback and take into account the comments in the review for the final version of the paper.